# Combining Crocin and Sorafenib Improves Their Tumor-Inhibiting Effects in a Rat Model of Diethylnitrosamine-Induced Cirrhotic-Hepatocellular Carcinoma

**DOI:** 10.3390/cancers15164063

**Published:** 2023-08-11

**Authors:** Basma Awad, Alaaeldin Ahmed Hamza, Amna Al-Maktoum, Suhail Al-Salam, Amr Amin

**Affiliations:** 1Biology Department, College of Science, United Arab Emirates University, Al Ain P.O. Box 15551, United Arab Emirates; basma.a.awad@hotmail.com (B.A.); 202032636@uaeu.ac.ae (A.A.-M.); 2National Organization for Drug Control and Research, Giza 12611, Egypt; alaa17mm@gmail.com; 3National Committee for Biochemistry and Molecular Biology and Medical Research Council, Academy of Scientific Research, Cairo 11334, Egypt; 4Department of Pathology, College of Medicine and Health Sciences, United Arab Emirates University, Al Ain P.O. Box 15551, United Arab Emirates; suhaila@uaeu.ac.ae

**Keywords:** hepatocellular carcinoma, crocin, sorafenib, inflammation, apoptosis

## Abstract

**Simple Summary:**

Liver cancer represents one of the most lethal forms of cancer, with hepatocellular carcinoma (HCC) accounting for the majority of its incidences and deaths. Currently, sorafenib is the first-in-line option for treating advanced and unresectable HCC. It is a multi-kinase inhibitor that intervenes with tumor growth and progression. Considering the modest results provided by sorafenib, identifying novel approaches to treating HCC remains a clinical imperative. Based on our previous work, crocin, a constituent of saffron, prevented HCC development. This study aimed to investigate its therapeutic effect in combination with sorafenib against an induced model of hepatocellular carcinoma arising from a cirrhotic milieu in rats. Our results confirmed that combination therapy yielded more effective outcomes compared to crocin and sorafenib monotherapies. It exerted the most pronounced effects in inhibiting inflammation and tumor cell proliferation while activating apoptosis and restoring macroscopic and cellular liver morphology. These results introduce a potential strategy for optimizing the anticancer effects of sorafenib using the bioactive natural compound crocin against HCC.

**Abstract:**

Hepatocellular carcinoma (HCC) is one of the most aggressive malignancies, with continuously increasing cases and fatalities. Diagnosis often occurs in the advanced stages, confining patients to systemic therapies such as sorafenib. Sorafenib (SB), a multi-kinase inhibitor, has not yet demonstrated sufficient efficacy against advanced HCC. There is a strong argument in favor of studying its use in combination with other medications to optimize the therapeutic results. According to our earlier work, crocin (CR), a key bioactive component of saffron, hinders HCC development and liver cancer stemness. In this study, we investigated the therapeutic use of CR or its combination with SB in a cirrhotic rat model of HCC and evaluated how effectively SB and CR inhibited tumor growth in this model. Diethylnitrosamine (DEN) was administered intraperitoneally to rats once a week for 15 weeks, leading to cirrhosis, and then 19 weeks later, leading to multifocal HCC. After 16 weeks of cancer induction, CR (200 mg/kg daily) and SB (10 mg/kg daily) were given orally to rats for three weeks, either separately or in combination. Consistently, the combination treatment considerably decreased the incidence of dyschromatic nodules, nodule multiplicity, and dysplastic nodules when compared to the HCC group of single therapies. Combined therapy also caused the highest degree of apoptosis, along with decreased proliferating and β-catenin levels in the tumor tissues. Additionally, when rats received combined therapy with CR, it showed anti-inflammatory characteristics where nuclear factor kappa B (NF-κB) and cyclooxygenase-2 (Cox-2) were considerably and additively lowered. As a result, CR potentiates the suppressive effects of SB on tumor growth and provides the opportunity to strengthen the therapeutic effects of SB in the treatment of HCC.

## 1. Introduction

Cancer has remained a pervasive health burden despite the advances in its prevention and treatment. It is the second leading cause of death worldwide, with almost 19.3 million cases and 10 million deaths reported in 2020 alone [1]. It has been estimated that cancer might soon potentially surpass heart disease as the leading cause of death [2]. Liver cancer represents the third deadliest cancer worldwide, with a predominance in Asian and African nations [3]. In the UAE, liver cancer ranks as the eighth most lethal cancer [4], with its incidence steadily on the rise [5]. Hepatocellular carcinoma (HCC) is the predominant primary liver cancer, constituting approximately 80% of all liver cancers [6]. Persistent and unresolved chronic inflammation from recurrent injuries to the liver leads to cirrhosis, fibrosis, and ultimately the growth of hepatic tumors [7]. Chronic infections with hepatitis B and C viruses, alcohol abuse, and exposure to environmental carcinogens like aflatoxins and nitrosamines are predisposing factors known to generate hepatic inflammation, ultimately leading to HCC [8,9,10]. Diethylnitrosamine (DEN), a substance found in tobacco smoke, processed foods, cosmetics, gasoline, and agricultural chemicals, is regarded as an environmental carcinogen to which constant exposure is unavoidable [11,12,13].

Advanced HCC has an extremely dismal prognosis, with over 50% of all HCC patients being diagnosed at a late stage [14]. Sorafenib (SB) is the primary approach for the treatment of advanced HCC and was approved by the U.S. Food and Drug Administration (FDA) in 2007 [15]. It is an oral multi-kinase inhibitor that targets Raf serine/threonine kinases, c Raf, wild-type, and mutant B Raf signaling to reduce tumor cell growth and promote tumor cell death. It prevents angiogenesis by inhibiting receptor tyrosine kinases, such as platelet-derived growth factor receptor (PDGFR) and vascular endothelial growth factor receptor (VEGFR) 2 and 3 [16,17]. Despite being the first medication to provide a prognostic advantage in advanced HCC, SB’s efficacy is very moderate, with a key phase III trial finding that the median overall survival was 10.7 months compared to 7.9 months with a placebo [18]. Additionally, persistent SB use frequently causes tumor cells to lose sensitivity, developing acquired resistance [19]. Combination therapy that targets several signaling pathways may be superior to monotherapy because it may be able to overcome resistance, feedback activation, and compensatory activation of pro-survival pathways [19]. Thus, combining SB with additional anticancer drugs may offer a novel therapeutic approach that could enhance anti-tumor effectiveness and circumvent SB resistance in HCC [15,20].

Historically, nature has provided a dependable and abundant supply of effective anticancer compounds with minimal to nonexistent negative effects [21]. Important anticancer agents such as Vinca alkaloids, podophyllotoxin, taxol, camptothecin, and their derivatives are among the 40% of FDA-approved medicinal chemicals that are of natural origin [21,22]. Many of these compounds are considered to be phytochemicals; therefore, plants have been extensively investigated in the search for potential anticancer agents against HCC [23]. Saffron (stigma of the *Crocus sativus* flower) and its major constituents have been explored for their anti-inflammatory, antioxidant, and anticancer properties [24,25,26]. For instance, saffron and most of its main components, such as crocin (CR), crocetin, and safranal, have been shown to significantly inhibit the growth of a variety of tumor cell lines, such as prostate, cervical, leukemia, lung, and liver cancer [25,26,27,28,29]. Saffron-derived CR also decreased the incidence of gastric cancer in methyl-3-nitro-1-nitrosoguanidine-induced rodents [30] and HCC in DEN-treated rats [25,26,31,32,33].

The environmental carcinogen DEN has the ability to generate lesions in rats resembling human benign and malignant tumors [12]. Therefore, it is widely employed to mimic and examine various human benign and malignant tumor types in laboratory settings [34]. Our previous study’s initiation-promotion model of cancer development was utilized to simulate the early stages of carcinogenesis [35]. In this investigation, a single dose of DEN, followed by 2-acetylaminofluorene, were intraperitoneally injected into rats to cause HCC. The study examined the therapeutic effects of CR alone or in conjunction with SB in rats during the early phases of hepatocellular tumor promotion. The combination of SB and CR more effectively lowered cell proliferation and carcinogenesis in this liver cancer model by reducing liver damage, oxidative stress, inflammation, and other adverse effects. To date, in vivo testing of SB and CR cotreatment in advanced HCC has yet to take place. In order to determine whether this therapeutic approach could improve the treatment of advanced HCC without causing greater toxicity, we coupled CR and SB in this study. In humans, approximately 80% of advanced HCC develops from cirrhotic livers [36]. Therefore, novel therapeutic approaches ought to be tested in an animal model of HCC occurring on a cirrhotic background. A rat model of DEN-induced liver injury that replicates the evolution of cirrhosis toward HCC was developed for the current study [37]. Multiple smaller doses of DEN are used in this procedure to cause liver cancer in animals. This is implemented to mimic the progression of liver cancer in HCC patients by simulating the inflammatory conditions from which fibrosis and cirrhosis arise before malignant tumors are formed [38]. This animal model with repeated-dose DEN is a useful model regimen for multifocal nodular HCC in the evaluation of novel combination therapy regimens [39]. To investigate the safety and antitumor effectiveness of the combination of SB with CR, we employed the DEN-induced cirrhotic rat model with HCC.

## 2. Materials and Methods

### 2.1. Materials

CR, DEN, Folin’s reagent, sodium pentobarbital, and bovine albumin were sourced from Sigma Chemical Co. (St. Louis, MO, USA). The alanine transaminase (ALT) (ab105134) and aspartate aminotransferase (AST) (ab105135) kits were purchased from Abcam (Cambridge, CB2 0AX, UK). Rabbit monoclonal antibodies against Bax [E63] (ab32503), β-catenin [E247] (ab32572,) COX2 [EPR12012] (ab179800), Ki67 antibody (ab15580), and NF-κB-p65 (ab16502) were purchased from Abcam (Cambridge, CB2 0AX, UK). Anti-proliferating cell nuclear antigen (PCNA), and anti-caspase-3 were purchased from Cell Signaling Technology Inc. (Danvers, MA, USA). Anti-caspase-9 was sourced from Novus Biologicals (Littleton, CO, USA), and anti-poly ADP-ribose polymerase (PARP), anti-Bcl-2, and anti-Bax for Western Blotting (WB) investigation were purchased from Santa Cruz Biotechnology (Santa Cruz, CA, USA). SB was purchased from Carbosynth Ltd. (Compton, UK).

### 2.2. Animals

A series of experiments were performed on 40 4-to-5-week-old healthy albino Wistar rats weighing 110–120 g. Upon protocol approval by the Animal Research and Ethics Committee of the College of Medicine and Health Sciences, UAE University (Approval No. A8-15), the rats were sourced from the college’s animal research facility. The animals were kept in a controlled environment with 12 h of light/dark cycles and a temperature range from 22 °C to 24 °C. Rats were habituated to the environment with unlimited access to tap water and a typical pellet diet.

### 2.3. Hepatocarcinogenesis Model

A model of cirrhosis-hepatocellular carcinoma was conducted in adherence to a modified version of the procedure outlined by DePeralta et al. [38] and Schiffer et al. [37]. The rats were separated into two groups at random: the control group and the experimental group. The control group (n = 8) received intraperitoneal injections of phosphate-buffered saline (PBS) once a week for 15 weeks. The experimental group (n = 32) received intraperitoneal injections of DEN at a dose of 50 mg/kg body weight on a weekly basis for 15 weeks, followed by seven days washout period to assess the development of HCC.

### 2.4. Experimental Design

Rats with established HCC were separated into four groups (n = 8) at the end of the 16-week HCC induction phase (HCC, HCC + SB, HCC + CR, HCC + CR + SB). In the control and HCC groups, 1 mL water containing 0.3% DMSO was administered orally five times each week for three weeks. The HCC-induced groups received treatments of CR and/or SB 5 times a week for 3 weeks. All drugs were administered orally via intragastric tubes. The HCC + SB group received SB dissolved in 0.3% DMSO at a dose of 10 mg/kg b.wt. The HCC + CR group received 200 mg/kg b.wt. of CR after it had been dissolved in 1 mL of water. The last HCC group was treated with both CR and SB. This group received 200 mg/kg b.wt. of CR with the subsequent administration of 10 mg/kg b.wt. of SB. The CR dose utilized in this study was determined based on previous studies assessing its toxicity and anti-inflammatory and anticancer benefits in rats with DEN-induced liver cancer [26,40]. Similarly, rats were given SB at a dose of 10 mg/kg based on findings on its therapeutic effect on the same cirrhotic rat model of hepatocellular carcinoma [19,41]. By the end of the experiment and after a 24 h post-last medication delivery, animals were euthanized by cervical dislocation under 3% sodium pentobarbital (45 mg/kg, i.p.) and dissected under controlled conditions. The experiment’s layout is shown in (Figure 1).

### 2.5. Sample Preparation

After 19 weeks of treatment with DEN, blood was collected from the retro-orbital plexus of rats under the effect of diethyl ether 24 h post-last treatment. The rats were then euthanized following the administration of 3% sodium pentobarbital (45 mg/kg, i.p.) by a cervical dislocation. The blood samples were centrifuged at 3000 rpm for 20 min (4 °C) for serum separation and collection. Gross livers were dehydrated, rinsed off with an ice-cold saline solution, and then dried out with blotting paper. A portion of the liver was immediately fixed in 10% buffered formalin for histological and immunohistochemical analyses. The remaining parts were flash-frozen in liquid nitrogen and kept at −80 °C for western blotting.

### 2.6. Liver Gross Appearance and Histopathological Analysis

After liver collection, macroscopic characteristics such as liver color, size, texture, and number of nodules were documented. The dyschromatic nodules were recognizable by their grayish-white color. These surface nodules exceeded 3 mm in size and were counted by two independent investigators. Samples collected from the right, left, and caudate lobes were fixed in 10% buffered formalin for histopathological examination. Fixed tissue samples representing the different groups (control, HCC, HCC + SB, HCC + CR, and HCC + CR + SB) were processed simultaneously and embedded in the paraffin block. Tissue blocks were serially cut into 5 μm thick sections, stained with hematoxylin and eosin (H&E), and then examined using an Olympus BX41 microscope (Tokyo, Japan). An experienced pathologist assigned the histopathological ratings. The percentage of the area covered by foci of cytoplasmic vacuolization or hydropic degeneration was used to grade hepatic vacuolization and hydropic degeneration: 0 corresponded to the range of 0–5%, 1 to 33%, 2 to 66%, and 3 to 66% [42]. The degree of inflammation was measured as the number of foci per 200-field: A focus was defined as zero, one to two, two to four, or three to four. The degree of fibrosis was rated as follows: 0 = no fibrosis, 1 = mild fibrosis (collagen fibers extend from the portal triad or central vein to the peripheral region), 2 = moderate fibrosis (collagen fibers create a fibrous septum without compartment development), and 3 = severe fibrosis (thick fibrous septum accompanied by pseudo-lobe formation). Large hepatocytes with hyperchromatic nuclei and dysplastic nodules (DN) were graded according to the proportion of the area they covered: 0 meant no DN, 1 meant 1%–25%, 2 meant 25%–50%, 3 meant 50%–75%, and 4 meant 75%–100% [42].

### 2.7. Enzyme Markers of Liver Injury

Using a commercially available colorimetric test, the liver enzyme activities of ALT and AST were measured. The Promega GloMax microplate reader was used to measure the concentrations of ALT (ab105134) and AST (ab105135) using the kits that were acquired from Abcam.

### 2.8. Immunohistochemical Staining

Mounted sections were submerged in sodium citrate buffer (0.1 M, pH 6) and left in a water bath for 15 min to expose antigen epitopes. This was followed by the incubation of the sections with 0.3% H_2_O_2_ in methanol to prevent any nonspecific binding to endogenous peroxidase. Rabbit anti-rat primary antibodies were added to the sections and incubated overnight at 4 °C. Rabbit monoclonal antibodies against Bax [E63] (ab32503), β-catenin [E247] (ab32572,) COX2 [EPR12012] (ab179800), Ki67 antibody (ab15580), and NF-κB-p65 (ab16502) were purchased from Abcam (Cambridge, CB2 0AX, UK), while anti-PCNA and anti-caspase-3 were purchased from Cell Signaling Technology Inc. (Danvers, MA, USA). Next, the slides were rinsed with PBS and subjected to incubation with the secondary antibody polyvalent biotinylated goat-anti-rabbit for 10 min at room temperature (1:200 dilution). The standard staining protocol was conducted using the Universal LSAB kit and DAB plus substrate kit. Additional counter-staining was carried out using hematoxylin. The slides were examined and photographed under an Olympus DP71 optical microscope. Ten fields were then chosen at random for the quantification of positive cells in individual samples (×400).

### 2.9. Quantification of Protein Expression in Tissue Sections

The quantitative assessment of different HCC biomarkers expressed in liver tissue sections was achieved by producing digital images using Image-Pro software version 6.0 for Windows. The percentage (%) of positively stained cells was detected by the ImageJ analysis program (NIH, Bethesda, MD, USA). Positively stained cells for Ki-67 and PCNA were counted belonging to five different fields of ten different sections of the different groups. The percentage (%) of the area of immunostaining per image was recorded at a final magnification of ×400. In total, 10 images per probed tissue section were examined corresponding to each group of rats.

### 2.10. Western Blotting

Cold RIPA buffer containing 2 μL of protease inhibitor and phosphatase inhibitor was utilized to homogenize and lyse the cells of 10 mg liver tissue. The separation of lysates was accomplished by centrifuging the samples at 16,000 rpm for 15 min at 4 °C. Heated samples (20 μg) in Laemmli buffer containing 4% SDS, 10% 2-mercaptoethanol, 20% glycerol, 0.004% bromophenol blue, and 0.125 M Tris HCl were employed to denature proteins in the liver homogenate. The pH was measured and maintained at 6.8. The samples were then heated to 95 degrees for 5 min. The Bradford method (Bio-Rad, Hercules, CA, USA) was used to determine protein concentrations. Protein samples of equal proportions were electrophoresed. Sodium dodecyl sulfate-polyacrylamide gel electrophoresis (SDS-PAGE) was carried out as directed by the manufacturer using a TGX Stain-Free Fast Cast Acrylamide Kit from Bio-Rad Laboratories Inc. (Hercules, CA, USA). Proteins were deposited onto polyvinylidene difluoride membranes (Millipore Bedford, MA) and blocked in bovine serum albumin (BSA, 3%) and Tris buffered saline/tween 20 (TBST) at room temperature for one hour after each sample’s protein content was separated on the SDS-PAGE gel. The membranes were treated with primary antibodies for an overnight period at 4 °C following the blocking stage. Anti-PCNA, anti-Caspase-3 (Cell Signaling Technology Inc., (Danvers, MA, USA), anti-caspase-9 (Novus Biologicals, Littleton, CO, USA), anti-poly ADP-ribose polymerase (PARP), anti-Bcl-2, and anti-Bax (Santa Cruz Biotechnology, Santa Cruz, CA, USA) were the main antibodies used for this investigation. After incubating the membranes with primary antibodies overnight, they were thoroughly cleaned with TBST before being re-probed for an hour at room temperature with secondary antibodies such as anti-rabbit IgG and anti-mouse IgG from Cell Signaling Technology, Inc. (Danvers, MA, USA). The signal was then detected and seen using the Bio-Rad ChemiDoc XRS+ System. Protein bands were then recognized using a chemiluminescence solution known as WesternSure PREMIUM. The ImageJ program was used to measure the band’s intensity. Considering the upregulation of common housekeeping proteins in HCC, using the amount of total protein was a better alternative for studies of protein expression that require normalization [43,44]. Total protein was utilized as the internal control and was stacked into a single band on a 5% SDS-PAGE stacking gel with no separating gels. SYPRO Ruby protein gel stain was utilized for total protein staining in accordance with the manufacturer’s (#170-3125, BIO-RAD, Hercules, CA, USA) protocol [44]. The band intensities were quantified using ImageJ software (National Institutes of Health).

### 2.11. Statistical Analysis

One-way analysis of variance (ANOVA) of the resulting data was conducted with SPSS statistical program version 25 (SPSS Inc., Chicago, IL, USA). Tukey’s post hoc analysis (*p* < 0.05) was carried out to investigate the differences between the means of the treated groups following the detection of significant differences by ANOVA.

## 3. Results

### 3.1. CR, SB, and Their Combination Inhibit Nodule Formation and Restore Liver Function of DEN-Induced HCC in Rats

Normal liver morphology and function were almost fully restored following adjuvant therapy with CR and SB in DEN-induced HCC. When observed with the naked eye, gross livers from the untreated HCC group were characterized by having an overall pale color and an ample number of grayish-white nodules (Figure 2A–C). This effect was alleviated to some degree in all the treated groups. SB administration (HCC + SB) markedly improved the macroscopic morphological status of the liver. CR-treated animals (HCC + CR) had a significantly lower number and multiplicity of nodules compared to the livers of animals treated with SB alone (HCC + SB) (Table 1 & Figure 2A,B). The therapeutic administration of both CR and SB (HCC + CR + SB) resulted in the highest level of liver resemblance to the control group. This group presented a glossy dark brown color of the liver and had the lowest nodule multiplicity and incidence (Table 1 & Figure 2A,B). Some rats treated with the combination regimen lacked any macroscopic nodules (Table 1). Biochemical serum markers ALT and AST were measured to assess the severity of liver damage in each of the experimental groups. Hepatic damage caused by DEN administration was indicated by a significant rise in the ALT and AST levels when compared to the control group. Treatment with SB, CR, and CR + SB attenuated the hepatic damage induced by DEN and notably lowered the serum activities of ALT and AST (Figure 2C,D). Co-therapy intensified the effects of SB, leading to a larger reduction in AST and ALT as opposed to the use of SB in isolation (Figure 2D).

### 3.2. CR, SB, and Their Combination Improve Histological Changes and Decrease Dysplastic Nodule Formation in Livers of HCC-Induced Rats

Histopathological analyses of liver tissue were carried out to inspect the role of CR, SB, and CR + SB in restoring normal cell and tissue morphology. Microscopic examinations of H&E-stained liver sections from the control group displayed proper hepatic architecture. This was characterized by hepatocytes arranged in a polygonal manner, radiating outward from the central vein of each lobule. The lobules had the usual thickness, and the hepatic sinusoids were intact. The hepatocytes possessed 1–2 dark blue nuclei with distinguishable nucleoli (Figure 3A,B). DEN treatment resulted in severe morphological deformations and a collapse of the normal polygonal hepatic architecture. Multi-dysplastic nodules (DN) with proliferating oval cells were observed in the HCC group (Figure 3C). These regions contained irregular dysplastic hepatocytes exhibiting cellular features such as nuclear hyperchromatism, karyomegaly, increased nucleocytoplasmic (N/C) ratios, and multiple nucleoli and nuclei (Figure 3D–F). Additionally, DEN administration generated areas of hydropic degeneration and trabecular HCC. All experimental HCC groups were associated with varying degrees of histological abnormalities. The SB group retained some DEN-induced cellular features associated with foci of hydropic degeneration (HD) (Figure 3G). However, SB clearly improved the histological structure when compared to the HCC group (Figure 3H). Likewise, CR administration demonstrated a decline in the foci of hydropic degeneration (HD) (Figure 3I,J). Cotreatment with CR and SB surpassed the efficacy of either treatment alone in improving the histological conditions and reversing the DEN-induced hepatocyte plate disorganization (Figure 3K,L). The degeneration of foci of hydropic degeneration along with the presence of some hepatocytes with pyknotic nuclei was observed in this group (Figure 3L). As shown in Figure 4, all the treated groups enhanced the histopathological scores for (A) foci of hydropic degeneration, (B) vacuolization, (C) inflammation, (D) fibrosis, and hyperchromatic nuclei with high N/C ratios. However, combined therapy with CR and SB demonstrated the highest impact in improving such scores.

### 3.3. CR Enhances Antiproliferative Effect of SB and Additively Downregulates β-Catenin Expression in HCC-Induced Rats

Proliferative markers were compared in the liver sections from all the groups. The nuclear proteins Ki-67 and PCNA are established indicators of aberrant cell proliferation that often coincide with tumorigenesis [45]. Liver sections from the HCC group displayed a significantly larger number of Ki-67- and PCNA-positive cells relative to the control group, as detected by IHC staining. After treatment with CR or SB individually, the number of proliferating cells significantly decreased. CR combined with SB exerted an additive effect on hindering cell proliferation and restored Ki-67 and PCNA levels in HCC tissues almost to that of the control group (Figure 5A–C). This effect was further verified by Western blotting, in which PCNA protein levels were markedly lowered in the groups treated with SB, CR, and CR + SB when compared to the HCC group (Figure 5D,E). Combination therapy caused the most dramatic reduction of PCNA, achieving expression levels lower than that of the control group (Figure 5D,E). The overexpression of β-catenin has been shown to be associated with hepatocarcinogenesis [46,47]. This was substantiated through IHC staining of β-catenin in which its expression was considerably higher in the HCC group (Figure 5F–H). The assessment of the number of positively stained cells showed that the HCC group animals had higher levels of cytonuclear expression of β-catenin (Figure 5H). Tissues obtained from the rats treated with SB or CR demonstrated mild β-catenin IHC reactivity. Cotreatment provided the most potent effect, as indicated by β-catenin’s intense suppression when compared to the HCC and control groups.

### 3.4. CR Promotes Apoptotic Effect of SB in HCC-Induced Rats

The pro-apoptotic efficacy of CR, SB, and CR + SB against HCC was evident via IHC staining and Western blot analyses measuring the activity of key apoptotic proteins (Figure 6). Pro-apoptotic proteins demonstrated an overall upregulation in all the treated groups when compared to the HCC group. The number of caspase 3- and Bax-positive cells was markedly increased in these groups (Figure 6A–C). However, this elevation was particularly prominent in the group treated with both CR and SB (HCC + CR + SB) (Figure 6A–C). This was corroborated by Western blot results in which the expression of Bax was increased in the SB and CR groups but more significantly in the CR + SB group (Figure 6D,F). This pro-apoptotic pattern was consistent with the downregulation of anti-apoptotic protein Bcl-2 in all the treated groups, especially when CR was combined with SB (Figure 6D,G). Similarly, a considerable reduction of PARP and caspase precursors, pro-caspase 3 and pro-caspase 9, was observed in the CR and SB-treated groups (Figure 6D,E,H,I), indicating the intrinsic activation of apoptosis. The most pronounced reduction was observed in the CR + SB group (Figure 6D,E,H,I). These results reflect CR’s role in strengthening the apoptotic capacity of SB against HCC.

### 3.5. CR Augments the Anti-Inflammatory Capacity of SB in HCC-Induced Rats

Increased expression of the inflammatory proteins COX-2 and NF-κB-p65 was exhibited in the HCC group. These proteins were especially concentrated around the central vein and in the Kupffer cells. There was a substantial decline in NF-κB-p65 and COX-2 immuno-positive cells following treatment with CR, SB, and CR + SB, indicating the inhibition of NF-κB-p65 activation and consequently a downregulation in inflammation. The anti-inflammatory effect of SB was amplified when it was combined with CR, as shown in Figure 7.

## 4. Discussion

The multi-kinase inhibitor SB targets key hallmark pathways of HCC and has long been the standard therapy for treating advanced HCC as a single agent [15,20]. However, drug resistance, unsatisfactory patient performance, and adverse drug-related side effects continue to impede the success of orally administered SB [18,19]. It has been reported that SB is only effective in about 30% of HCC patients who eventually gain resistance within 6 months [48]. This highlights the urgent need for discovering better therapeutic modalities that tackle SB-related challenges. Employing a combination regimen seems to be a sound strategy for increasing tumor sensitivity to the drug [15,19]. However, increased toxicity is a major drawback associated with this approach. Therefore, identifying well-tolerated and efficient combinations of targeted therapy is necessary for the treatment of HCC patients. In this study, a cirrhotic rat model of HCC was adopted to evaluate the interaction of SB with CR. The progression of HCC from cirrhosis is observed in around 90% of HCC patients, regardless of the etiology of liver disease [49]. This indicates that the animal model utilized in this study accurately reflects the physiopathology of human HCC [38,39]. The amalgamation of SB with CR was shown to offer greater anticancer results against HCC in a synergistic manner when compared to SB and CR monotherapies in rats. Combination therapy caused the most dramatic mitigation of tumor incidence, cell proliferation, and inflammation while promoting apoptosis in DEN-induced HCC.

In the current investigation, dichromatic tumor nodules were significantly more prevalent on the surface of the livers of cirrhotic rats with HCC. The histological study proved that these dichromatic tumor nodules were HCC. DN featured enlarged hyperchromatic nuclei with clear, eosinophilic, or hyper basophilic cytoplasm. These histological characteristics coexisted with other abnormalities such as steatosis, fibrosis, inflammation, hydropic degeneration, and cell necrosis. Under the circumstances of our experiment, administration of CR and SB to DEN-treated rats resulted in a considerable decrease in the number and incidence of dyschromatic nodules as well as a decline in FAH and other histological abnormalities. In terms of the quantity and occurrence of dyschromatic nodules as well as a reduction in FAH with other histological abnormalities, we discovered that the combination of CR and SB outperformed SB and CR monotherapies. These results confirm what was concluded in our recent study on the early stages of carcinogenesis, which proved that the combination of SB and CR increased the effectiveness of SB in lowering cell proliferation and carcinogenesis in this liver cancer model [35].

The status of liver function is often identified by assessing the ALT and AST levels in blood serum. In instances where hepatocytes are subjected to damage, these enzymes leak into circulation and are consequently markers for liver damage. Increased levels of these biochemical parameters often correlate with the incidence of HCC [50]. In this study, the elevated levels of ALT and AST in the HCC group represented the progression of carcinogenesis and showed that liver damage occurred when preneoplastic lesions first appeared. SB or CR monotherapy considerably alleviated the HCC-associated liver injury. These findings align with previous studies on CR’s ability in restoring ALT and AST levels in several different hepatotoxic conditions [51,52,53]. According to the results, the improvement of liver function following SB and CR co-administration was superior to CR or SB monotherapies. These benefits were demonstrated by a notable drop in ALT and AST activity, an indication of diminishing liver damage that may have been attributed to CR’s capacity to avoid liver damage and enhance anticancer activity. Our data is consistent with earlier investigations where the combination of anticancer compounds with SB augmented SB’s ability in decreasing the ALT and AST levels [54,55].

The characteristic pathways of HCC in cancer cells that promote tumor formation and progression include sustained proliferative signaling and the obstruction of apoptosis [56,57]. In this work, Ki-67 and PCNA were utilized to assess cellular proliferation, as they are exclusively expressed during that process [58]. All HCC animals treated with mono- or combined therapy showed the clear antiproliferative impact of CR and SB, where expressions of Ki-67 and PCNA were markedly downregulated. These results are in line with published research documenting the antiproliferative effects of CR in the liver cancer model induced by DEN [19,26,33,40]. The antiproliferative effects of SB and CR combined exceeded those imposed by CR and SB monotherapies. Increasing the effectiveness of the combination of SB and CR against liver cancer cell proliferation in this model is consistent with the early stages of carcinogenesis in our recent study [35].

Modulation of apoptosis plays a significant role in promoting carcinogenesis and conferring chemoresistance to cancer cells as they evade apoptosis [59,60]. Here, our HCC group demonstrated low Bax levels, whereas Bcl-2, pro-caspase-3, pro-caspase-9, and PARP levels rose, which is consistent with earlier studies by Thomas et al. [61] and Younis et al. [62]. Numerous cancer types have elevated levels of PARP, pro-caspase-3, and pro-caspase-9, and these aberrant expressions indicate that these markers play important roles in the development of oncogenic transformation [63,64]. According to the results presented here, treatment with CR, SB alone, or their combination promoted apoptosis in HCC liver tissue, as shown by the elevated levels of the pro-apoptotic proteins Bax and caspase-3 and the decreased levels of Bcl-2, PARP, pro-caspase-3, and pro-caspase-9 in comparison to the HCC animals. Furthermore, visual examination of the liver samples from various treatment groups confirmed the occurrence of hepatic vacuolation and pyknotic nuclei in cells that may undergo apoptosis, as well as the occurrence of regions displaying cytoplasmic shrinkage and condensation that might be related to apoptotic cell death. Numerous studies have linked CR and SB’s ability to inhibit tumor growth in the DEN animal model of cancer to their ability to induce apoptosis by upregulating caspase-3 and Bax and downregulating the anti-apoptotic protein Bcl-2, which in turn encourages apoptosis [33,40,41,62]. Compared to the individual effects, the combined effect was more robust. This is consistent with previous studies that looked at SB combination therapies to reduce SB resistance [65,66]. These combination therapies using natural compounds for HCC increased Bax and caspase-3 and decreased Bcl-2, indicating apoptotic activation to overcome SB resistance.

Treatment with CR, SB, and their combination was highly effective in reducing liver β-catenin overexpression as well as liver tumor and liver proliferation. According to several studies, up to 78% of HCCs have β-catenin activation in their tissue [46,47]. HCC, among other liver cancers, is exacerbated by β-catenin signaling [47,67], which has been linked to an increased β-catenin expression level in DEN-treated rats [68,69]. As a result of β-catenin activation and subsequent trafficking into the nuclei, downstream target genes are stimulated to express themselves, which causes aberrant cell proliferation and ultimately carcinogenesis [67]. Our findings imply that CR, SB, and their combination affected the β-catenin signaling system because they reduced the elevation of β-catenin protein levels in the HCC model. Therefore, by controlling the β-catenin signaling system, treatment with SB and CR reduced HCC cell proliferation and tumorigenicity. The downregulation of the β-catenin effects in the HCC model was shown to be improved by SB and CR combination over CR and SB alone. Here, we demonstrated how CR–SB combination therapy contributed to the additive downregulation of β-catenin tissue expression.

Chronic Inflammation marked by an overexpression of COX-2 has been linked to the development of HCC [70,71]. The rate-limiting enzyme, COX-2, is responsible for mitogenic stimuli and inflammation, which promote the synthesis of prostaglandins in neoplastic tissue and inflamed areas [70,71]. In the current investigation, the rats given DEN had much higher levels of COX-2 than the rats given a control, indicating that this model may be stimulating the inflammatory pathways. Our findings are consistent with previous reports that showed DEN treatment increased COX-2 levels [70,71]. When CR, SB, or both were administered, the inflammatory pathway was markedly suppressed, as evidenced by a substantial drop in the COX-2 levels when compared to rats who had received DEN treatment. In this respect, our findings are consistent with earlier investigations demonstrating the comparable effects of CR and SB on COX-2 [26,40]. The oncogenic transcription factor NF-κB regulates COX-2 in a variety of cancer cell types and mediates cross-talk between inflammation and cancer on a number of different levels [72]. Notably, numerous studies have demonstrated that NF-κB activation encourages the development of cancer. NF-κB promotes angiogenesis, invasion, metastasis, anti-apoptosis, and proliferation [72,73]. The present findings of NF-κB-p65 in the livers of DEN-treated rats imply that inflammation may have taken place as a result of its up-regulation [31]. The up-regulation of NF-κB-p65 in the HCC group was required for the activation of COX-2 and, consequently, inflammation and carcinogenesis, as further corroborated by our findings of NF-κB-p65 in the liver of DEN-induced rats. When comparing HCC-induced rats to those given CR, SB, or both, the level of NF-κB-p65 was dramatically lowered in all the treatments, and the upregulation of NF-κB-p65 levels was additively decreased by the CR–SB combination. In this sense, our findings are consistent with those of prior studies that found CR and SB had comparable effects on NF-κB-p65 levels [26,35,74,75].

## 5. Conclusions

In conclusion, our findings in the cirrhotic rat model of HCC suggest that CR, when combined with SB, potentiates the anticancer effects of SB. CR promotes apoptosis, and reduces cellular proliferation, making it a viable medication for the treatment of HCC with few side effects. The downregulation of NF-κB-p65, COX-2, and β-catenin is part of the mechanism behind these positive effects (Figure 8). Therefore, both in monotherapy and in combination therapy with already available treatments like SB, this inexpensive agent is a great candidate for pharmacological repurposing as an oncological treatment. These findings point to CR and SB as potential options for further clinical studies in the treatment of HCC. However, it is important to consider the limitations associated with the experimental design presented in this study. Establishing an animal model that reflects the exact etiological complexity and heterogeneity of human HCC is a challenge. It has been reported that DEN may need to be administered for at least a year to effectively induce an HCC model that is equivalent to human HCC [76]. Despite this, DEN-induced HCC in animal models provide reliable insights into the benefits of the presented treatment regimen.

## Figures and Tables

**Figure 1 cancers-15-04063-f001:**
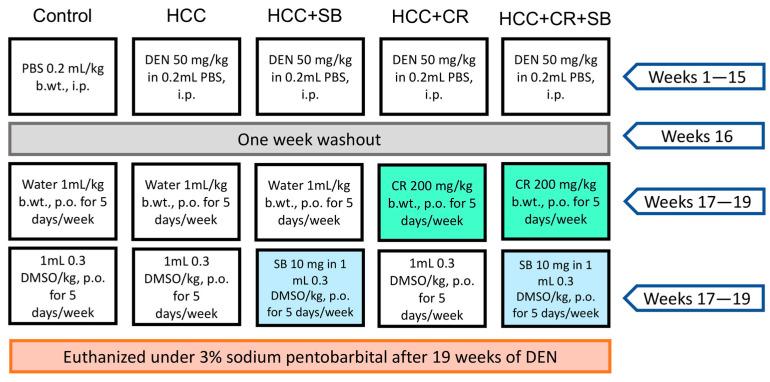
A visual representation of the experimental design. PBS: phosphate-buffered saline, DEN: diethyl nitrosamine, HCC: hepatocellular carcinoma, CR: crocin, SB: sorafenib, DMSO: dimethyl sulfoxide, i.p.: intraperitoneal, p.o.: per. os.

**Figure 2 cancers-15-04063-f002:**
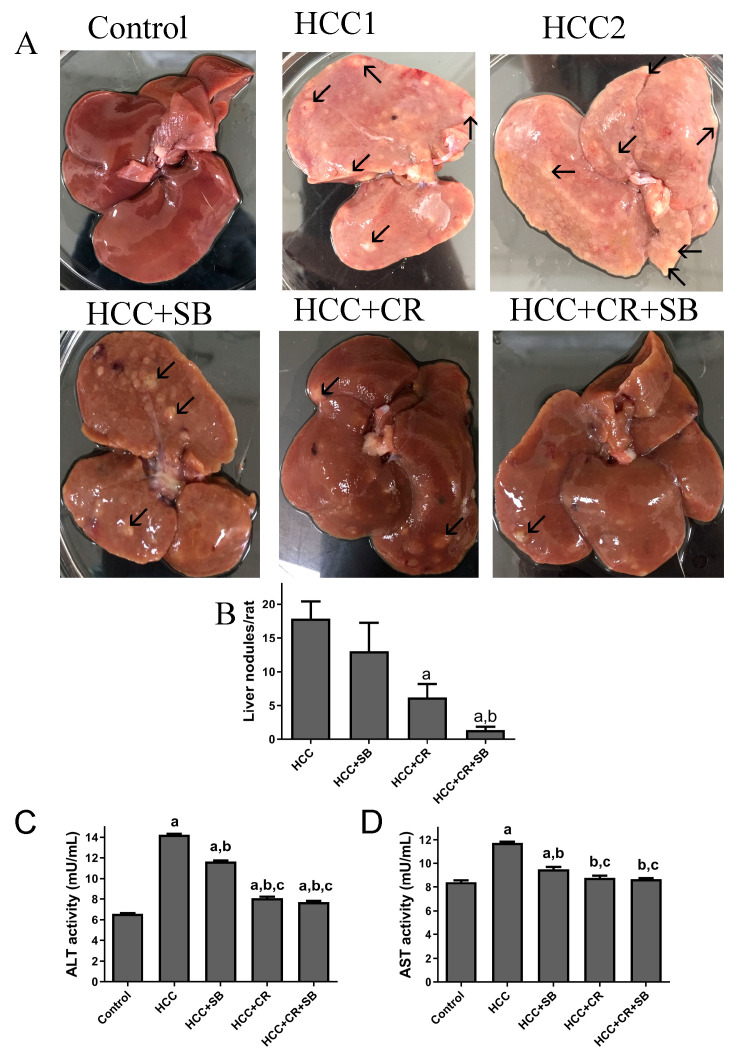
Gross investigations of rat livers: (**A**) Representative macroscopic appearances of livers of DEN-treated male rats administered with CR, SB, and CR + SB, arrows show liver nodules. (**B**) The liver nodule numbers/rat (>3 mm) were calculated. Values represent mean ± SEM of six rats per group. One-way analysis of variance followed by Tukey’s post hoc analysis was used to calculate significance values: a *p* < 0.05 vs. HCC group; b *p* < 0.05 vs. HCC + SB. (**C**,**D**) ALT and AST activity is expressed as mU/mL. One-way analysis of variance and a Dunnett’s *t* test were used to determine significance values: a *p* < 0.05 vs. control group; b *p* < 0.05 vs. HCC, c *p* < 0.05 vs. HCC + SB.

**Figure 3 cancers-15-04063-f003:**
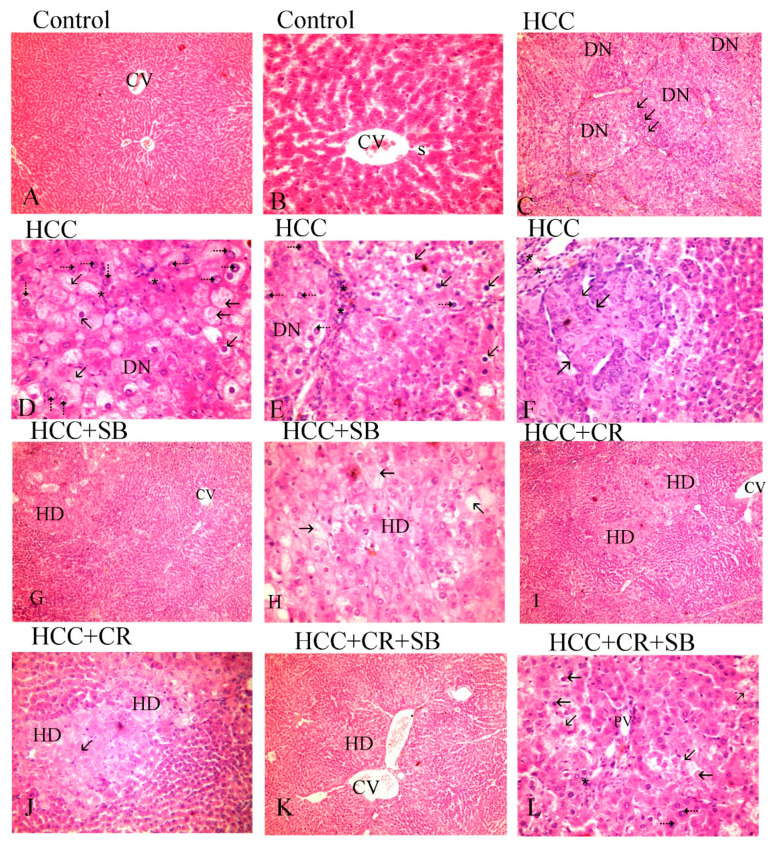
H&E-stained liver sections underwent histopathological analysis, with the control group (**A**,**B**), the HCC group (**C**–**F**), the HCC + SB group (**G**,**H**), and the HCC + CR + SB group (**K**,**L**). Histopathological investigations of liver sections stained with H&E. (**A**,**B**) control group, (**C**–**E**) HCC group, (**G**,**H**) HCC + SB group, (**I**,**J**) HCC + CR group, (**K**,**L**) HCC + CR + SB group. (**A**) Demonstrating proper hepatic organization characterized by hepatic lobules and oblique interlobular septa with polygonal hepatocytes extending outward from the central vein (CV) (×100). (**B**) The hepatocytes showed eosinophilic cytoplasm and vesicular nuclei. Flat endothelial cells and Kupffer cells bordered the sinusoidal gaps (S) between the hepatocytes (×400). (**C**) The HCC group displayed a disruption of the hepatic architecture with multi-dysplastic nodules (DN) encircled by proliferating oval cells (↑) (×100). (**D**) Hepatocytes in the HCC group exhibited hydropic degeneration, irregularity, dysplasia, pleomorphic cells with karyomegalic nuclei, conspicuous numerous nucleoli (⇡), and other hepatocytes displayed shrunken darkened nuclei and vacuolated cytoplasm (↑). (×400). (**E**) Some hepatocytes showed a high nucleocytoplasmic ratio and hyperchromatic nuclei (⇡), whereas others showed vacuolated cytoplasm, shrunken nuclei (↑), and mononuclear cell infiltration (*), (×400). (**F**) Displaying trabecular liver cancer (↑) and inflammation (*). (**G**) A focal hydropic degeneration (HD), (×100), was seen in the HCC + SB group. (**H**) Other samples displayed foci of degeneration with total loss of hepatocytes, a sparse residual population of pyknotic nuclei (↑), and an improvement in the nucleocytoplasmic ratio, (×400). (**I**,**J**) The histological structure of the HCC + CR group showed improvement with a decline foci of hydropic degeneration (HD) (↑). (**K**) Hepatocytes were neatly distributed in cords radiating from the central vein in the HCC + CR + SB group, indicating an improvement in normal liver architecture (CV). (×100). (**L**) Exhibited areas of degeneration with full loss of hepatocytes and a small number of pyknotic nuclei and numerous nucleoli (⇡), along with the infiltration of inflammatory cells (*) and an improvement in the nucleocytoplasmic ratio.

**Figure 4 cancers-15-04063-f004:**
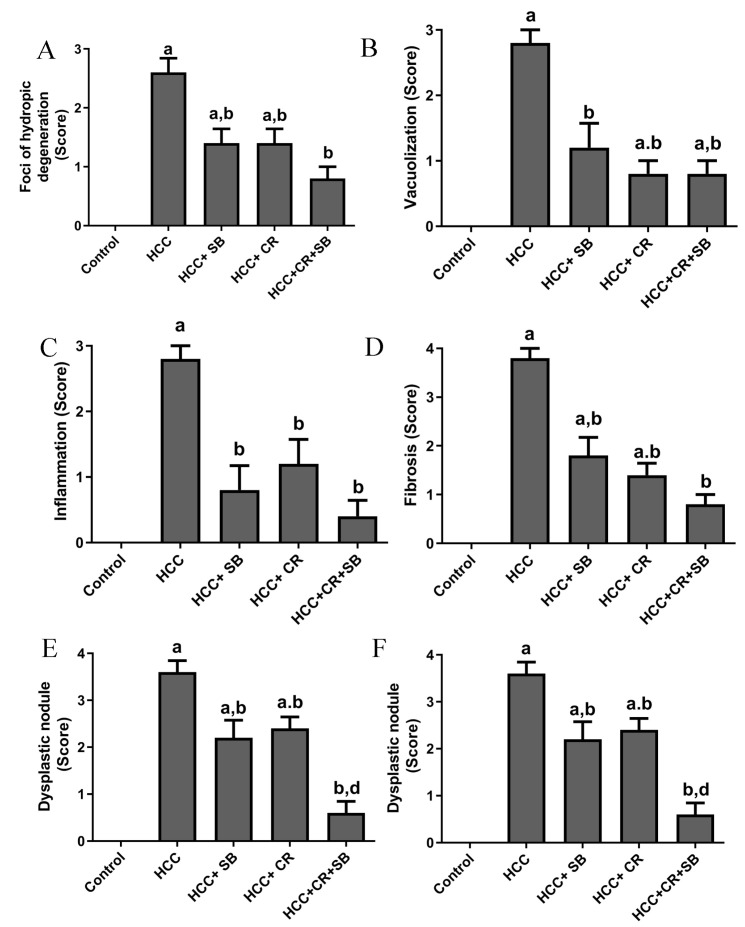
Histopathological scores for (**A**) foci of hydropic degeneration, (**B**) vacuolization, (**C**) inflammation, (**D**) fibrosis, and (**E**) hyperchromatic nuclei with high N/C ration, and (**F**) dysplastic nodules. Values expressed as mean ± SEM for five animals in each group. Significance was determined by one-way analysis of variance followed by Tukey’s post hoc analysis: a *p* < 0.05 vs. control group; b *p* < 0.05 vs. HCC group; d *p* < 0.05 vs. HCC + CR group.

**Figure 5 cancers-15-04063-f005:**
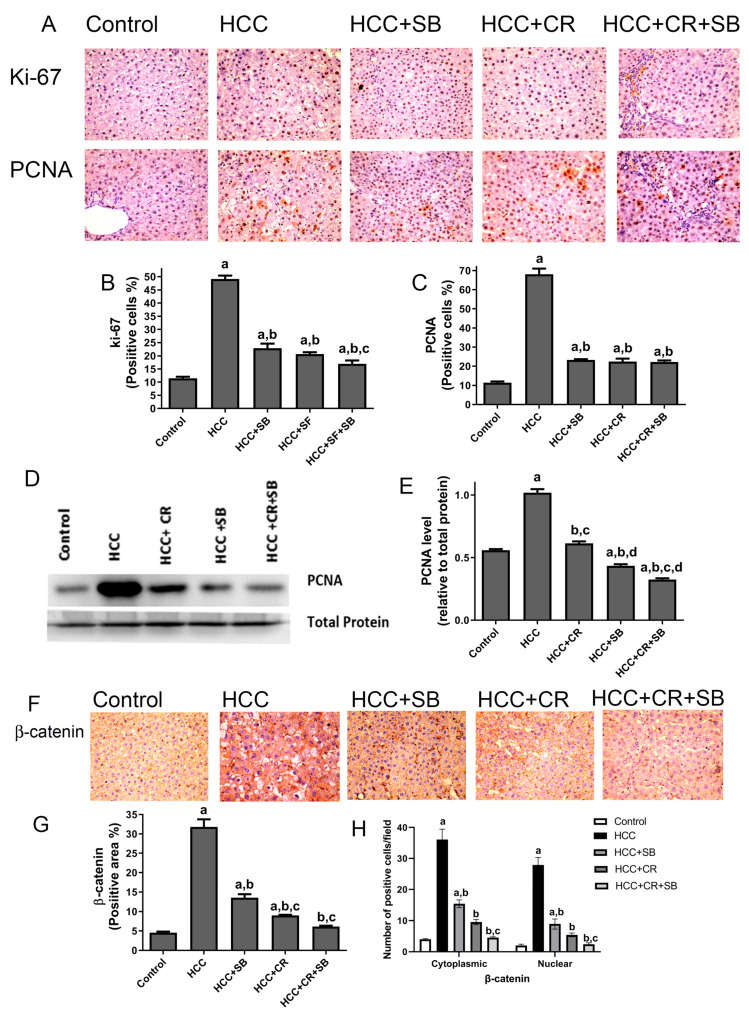
CR potentiates antiproliferative and β-catenin downregulation of SB-induced effects in HCC-induced rats. (**A**) IHC staining of Ki-67 and PCNA in rat liver tissue sections. (**B**,**C**) Graphs reflect the number of cells positively stained for Ki-67 and PCNA, respectively. Positive cell count was determined belonging to five different fields of ten different sections of different groups. (**D**) PCNA concentration in rat tissue lysate of each group quantified by WB. (**E**) Quantification plot illustrating WB signals quantified by ImageJ and normalized relative to the total protein from the liver. (**F**) β-catenin IHC staining in rat liver tissue sections. (**F**) An intense IHC reaction of β-catenin was observed in the HCC group when compared to the control group. Conversely, tissue sections obtained from rats treated with SB, CR, and CR + SB demonstrated a subtle IHC reaction of β-catenin when compared to the control and HCC groups. (**G**) A graphical representation of the IHC staining intensity of β-catenin corresponding to each group. (**H**) Graph showing the number of the positive cells for cytoplasmic and nuclear staining of β-catenin expression in each section by calculating the number in ten fields at ×400 magnifications and then the number of positive cells/fields. Values expressed as mean ± SEM of four animals in each group. The displayed data depicts three independent experiments performed in triplicates and expressed as mean values ± SEM (n = 4). a *p* < 0.05, were significant as compared with normal control rats and b *p* < 0.05 were significant as compared with HCC control group, c *p* < 0.05, were significant as compared with SB group, d *p* < 0.05 vs. HCC + CR group. Original western blots are Presented in Appendix A.

**Figure 6 cancers-15-04063-f006:**
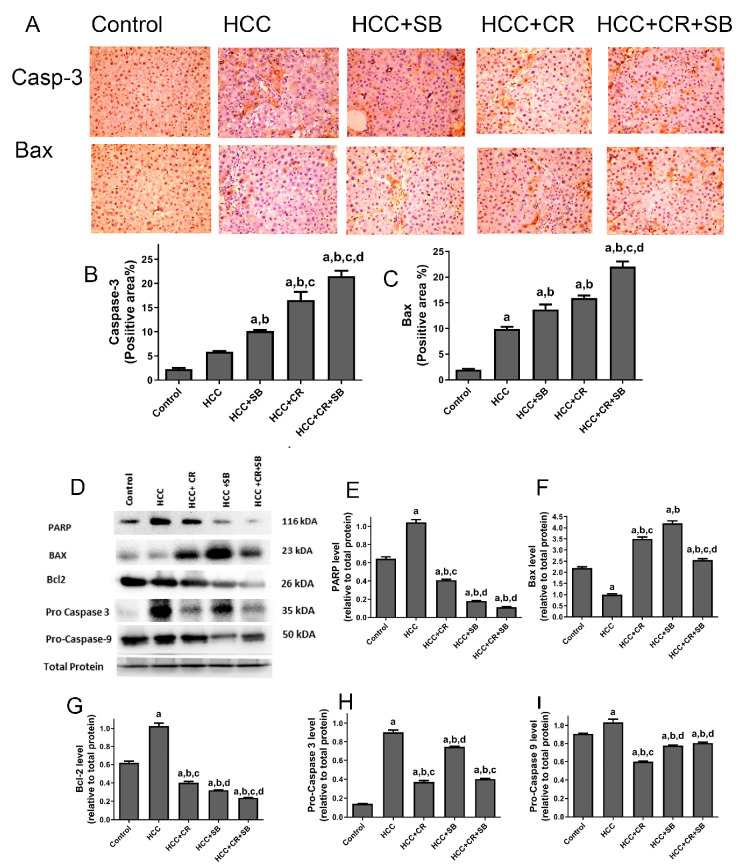
Effects of CR, SB, and CR + SB on apoptotic cell death. (**A**) Representative images exhibiting Bax and caspase 3 IHC staining of liver tissue sections corresponding to all groups (**B**,**C**) show the corresponding IHC-positive staining area graph of Bax and caspase 3-positive area in different experimental groups. The IHC staining of Bax and caspase 3 in each section was calculated by positive area of brown staining in ten fields at ×400 magnifications then the area of positive cells/field. (**D**) Western blot analysis showing the decreased expression of Bcl-2, Pro caspase-3, and Pro caspase-9, and the increased expression of Bax in SB, CR, and CR + SB treated groups. (**E**–**I**) show densitometric analysis representing the expression of PARP, Bax, Bcl-2, Pro caspase-3, and Pro caspase-9 normalized to that of total protein. Values expressed as mean ± SEM for six animals in each group. Significance was determined by one-way analysis of variance followed by Tukey’s post hoc analysis: a *p* < 0.05 vs. control group; b *p* < 0.05 vs. HCC group; c *p* < 0.05 vs. HCC + SB group; d *p* < 0.05 vs. HCC + CR group. Original western blots are Presented in Appendix A.

**Figure 7 cancers-15-04063-f007:**
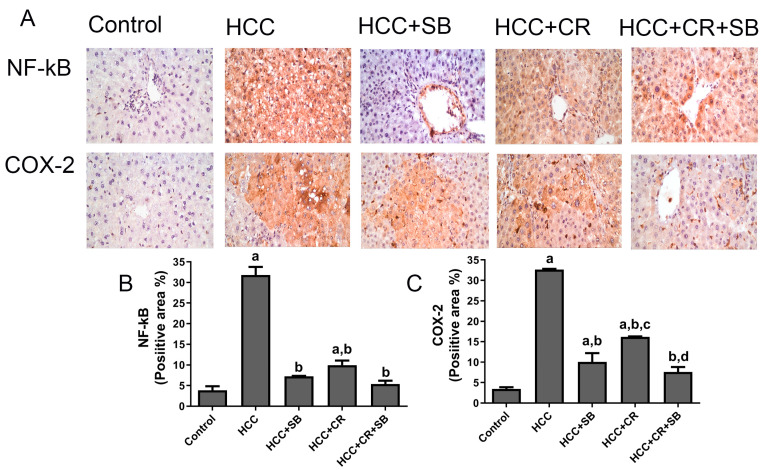
Inhibitory effect of CR, SB, and CR + SB on HCC-induced upregulation in NF-kB-p65 and COX-2 expression in liver. (**A**) Representative images of IHC NF-kB-p65 and COX-2 stained sections in the livers of all groups studied. (**B**,**C**) represent the quantification of COX-2 and NF-kB-p65-positive staining area. Increased NF-kB-p65 and COX-2 immunopositivity staining was observed in livers of the HCC group, as indicated by a brown color, as compared to the control group. The percent of positive staining area was reduced in SB, CR, and CR + SB treated groups. The expression of NF-kB-p65 and COX-2 in each section was determined by calculating the percent of positive area of brown staining in ten fields at ×400 magnifications and then the area of positive cells/field. Values expressed as mean ± SEM of six animals in each group. Significance was determined by one-way analysis of variance followed by Tukey’s post hoc analysis: a *p* < 0.05 vs. control group; b *p* < 0.05 vs. HCC group; c *p* < 0.05 vs. HCC + SB group, d *p* < 0.05 vs. HCC + CR group.

**Figure 8 cancers-15-04063-f008:**
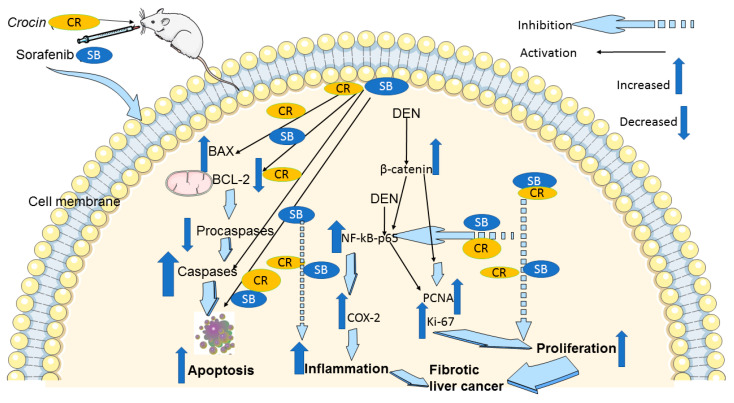
A schematic diagram of the anticancer molecular mechanism of CR and SB in cirrhotic liver cancer model. The pro-apoptotic proteins Bax and caspase-3 are upregulated by combination treatment CR and SB, while Bcl-2, pro-caspase-3, and pro-caspase-9 are downregulated. Thus, caspase-mediated apoptosis is induced. In addition, combination treatment reduces Ki-67 and PCNA proliferative markers. Reducing elevated protein levels of β-catenin and the inflammatory markers NF-κB and COX-2 in the HCC animal model leads to the antiproliferative and anti-inflammatory and apoptotic effects of combination therapy.

**Table 1 cancers-15-04063-t001:** Effects of CR, SB, and their combination on the incidence of cancer nodules.

Groups	Number of Rats with Nodules/Total Number of Rats	Nodule Incidence (%)	Inhibition of Nodule Incidence (%)	Total Number of Nodules	Mean Number of Nodules per Rat ^a^	Inhibition of Nodule Multiplicity (%)
HCC	6/6	100%	0%	107	17.83	0%
HCC + SB	6/6	100%	0%	78	13	27.1%
HCC + CR	6/6	100%	0%	37	6.17 ^b^	65.40%
HCC + CR + SB	4/6	66.67%	33.33%	8	2 ^b,c^	92.52%

^a^ Total number of nodules/numbers of animals bearing nodules in each group (multiplicity). ^b^ *p* < 0.05, as compared to HCC group, ^c^ *p* < 0.05 as compared to HCC + SB.

## Data Availability

All data are readily available in the manuscript.

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
