# Peer review of "Combining Crocin and Sorafenib Improves Their Tumor-Inhibiting Effects in a Rat Model of Diethylnitrosamine-Induced Cirrhotic-Hepatocellular Carcinoma"

_cancers, 2023, doi:10.3390/cancers15164063_

Round 1
Reviewer 1 Report
The manuscript reports that Sorafenib (SB) and Crocin (CR) as monotherapies, and Sorafenib/Crocin combination therapy can significantly reduce DEN-induced liver damage in rats. It may be considered a companion piece with a previous paper (Abdu et al., 2022), with which it shares one co-author. However, a sufficient amount of new data is presented to justify publication; the other paper is clearly acknowledged (reference [34])
Arguably, figures 2B and 2C refer to the same data. Of the two figures, 2B is easiest to understand (and has error bars, in contrast to 2C). In my opinion, figure 2C can be deleted.
Figure 3 reports the level magnification used for the figures in the caption. It would improve the manuscript if also scale bars were added in the figures to indicate the level magnification.
Figures 4A-F are quite rich in detail, making it rather difficult to immediately interpret the data. Five data points with their average value and a standard deviation bar, plus indication of significant differences (a, b). I would suggest leaving out the individual data points; just average, error bars, and significance indicators is sufficient.
Figures 4A-F are quite rich in detail, making it rather difficult to immediately interpret the data. Five data points with their average value and a standard deviation bar, plus indication of significant differences (a, b). I would suggest leaving out the individual data points; just average, error bars, and significance indicators is sufficient.
Minor corrections
Line
48 Cancer has remained a pervasive health burden...
266 Normal liver morphology and function were almost fully restored...
Author Response
REVIEWERS’ COMMENTS
Reviewer # 1
Comment 1:
The manuscript reports that Sorafenib (SB) and Crocin (CR) as monotherapies, and Sorafenib/Crocin combination therapy can significantly reduce DEN-induced liver damage in rats. It may be considered a companion piece with a previous paper (Abdu et al.,2022), with which it shares one co-author.
Response:
Our previous study (Abdu et al., 2022), which examined the therapeutic effects of CR alone or in conjunction with SB in rats, differs from this study and is not part of it for the following reasons:
1- It aimed to study the effect of CR in a different animal model where a single dose of DEN, followed by 2-acetylaminofluorene and that model differs from our model where multiple smaller doses of DEN were used to induce cirrhosis-hepatocellular carcinoma.
2- In the previous study, SB, CR and CR/sorafenib combination were given after seven weeks of cancer induction and for a period of six weeks. In this study, the drugs were given after sixteen weeks of cancer induction and for a period of three weeks.
3- Also, the doses of SB and CR were different in the two studies. In the first, SB was at a dose of 7.5 mg/kg, and in the newer one, it was 10 mg/kg. Also, the dose of CR was 50 mg/kg in the first and 200 mg/kg in the newer one.
Reference
Abdu, S.; Juaid, N.; Amin, A.; Moulay, M.; Miled, N., Therapeutic Effects of Crocin Alone or in Combination with Sorafenib against Hepatocellular Carcinoma: In Vivo & In Vitro Insights. Antioxidants (Basel) 2022, 11, (9).
Comment 2:
Arguably, figures 2B and 2C refer to the same data. Of the two figures, 2B is easiest to understand (and has error bars, in contrast to 2C). In my opinion, figure 2C can be deleted.
Response:
As suggested by the reviewer, we deleted figure 2C.
Comment 3:
Figure 3 reports the level magnification used for the figures in the caption. It would improve the manuscript if also scale bars were added in the figures to indicate the level magnification.
Response:
Now the level magnification used for the figures has been added, but we sincerely apologize for not being able to add scale bars in the figures due to a technical defect in the microscope program.
Comment 4:
Figures 4A-F are quite rich in detail, making it rather difficult to immediately interpret the data. Five data points with their average value and a standard deviation bar, plus indication of significant differences (a, b). I would suggest leaving out the individual data points; just average, error bars, and significance indicators is sufficient.
Response:
Based on your suggestion individual data points were excluded; The figure has been redrawn to contain only the mean, error bars, and significance indices.
Response to Minor corrections
As pointed out by the reviewer, the indicated phrase in line 51 is now revised, quote “Cancer has remained a pervasive health burden...”.
As pointed out by the reviewer, the indicated phrase in line 276 is now revised, quote “Normal liver morphology and function were almost fully restored”.
Reviewer 2 Report
This paper discusses on potential anticancer properties of Sorafenib and crocin in association in a rat experimental model of hepatocarcinoma. In the opinion of Authors, this association should potentiate the little impact of sorafenib alone. Above all considering that there are evidences that saffron and its carotenoids exert chemopreventive activity through anti-oxidant activity, cancer cells apoptosis, inhibition of cell proliferation, enhancement of cell differentiation, modulation of cell cycle progression and cell growth, modulation of tumor metabolism, stimulation of cell-to-cell communication and immune modulation.
The study is well realized (specifically, the adopted concentrations of drugs are right) even if the sample size is relatively low for the aims of this study as evidenced by statistical results.
Anyway, the study is interesting and stimulating and fit the scopes of the journal
The abstract is sufficiently informative and do gives a clear idea of the content of the paper.
The paper results well written and comprehensive. Just some little typing errors.
I would just suggest to better express their opinion and conclusions, considering the limits of this rat experimental model as well.
Moreover, I would suggest to better indicate the statistical significance for all groups studied.
The reference section is adequate.
This paper discusses on potential anticancer properties of Sorafenib and crocin in association in a rat experimental model of hepatocarcinoma. In the opinion of Authors, this association should potentiate the little impact of sorafenib alone. Above all considering that there are evidences that saffron and its carotenoids exert chemopreventive activity through anti-oxidant activity, cancer cells apoptosis, inhibition of cell proliferation, enhancement of cell differentiation, modulation of cell cycle progression and cell growth, modulation of tumor metabolism, stimulation of cell-to-cell communication and immune modulation.
The study is well realized (specifically, the adopted concentrations of drugs are right) even if the sample size is relatively low for the aims of this study as evidenced by statistical results.
Anyway, the study is interesting and stimulating and fit the scopes of the journal
The abstract is sufficiently informative and do gives a clear idea of the content of the paper.
The paper results well written and comprehensive. Just some little typing errors.
I would just suggest to better express their opinion and conclusions, considering the limits of this rat experimental model as well.
Moreover, I would suggest to better indicate the statistical significance for all groups studied.
The reference section is adequate.
Author Response
Reviewer # 2
Comment 1:
The study is well realized (specifically, the adopted concentrations of drugs are right) even if the sample size is relatively low for the aims of this study as evidenced by statistical results. Anyway, the study is interesting and stimulating and fit the scopes of the journal. The abstract is sufficiently informative and do gives a clear idea of the content of the paper. The paper results well written and comprehensive. Just some little typing errors.
Response:
Thank you very much for your wonderful evaluation of the research and your encouraging words. We appreciate and respect this evaluation.
Comment 2:
The paper results well written and comprehensive. Just some little typing errors.
Response:
Thank you very much. We have revised the English language and corrected typing mistakes.
Comment 3:
I would just suggest to better express their opinion and conclusions, considering the limits of this rat experimental model as well.
Response:
Thank you very much. We have now better expressed our opinions and conclusions, keeping in mind the limitations of this experimental rat model as well (Page 17, lines 578-591).
Comment 4:
Moreover, I would suggest to better indicate the statistical significance for all groups studied.
Response:
Thank you very much for your insightful observation and valuable suggestion. Now the statistical analysis has been re-analyzed to include all total groups, Tukey’s post hoc analysis (P < 0.05) was carried out to investigate the differences between the means of the treated groups following the detection of significant differences by ANOVA. All figures have been updated based on this new data
Reviewer 3 Report
Review comments
The article is interesting since it deals with hot topic which is the treatment of hepatocellular carcinoma with sorafenib in combination with a natural product, crocin. I recommend the acceptance of the article for publication after responding to the following comments:
1- Diethylnitrosamine is used in this model of cancer for initiation of carcinogenesis not to induce liver cirrhosis. Thus, the title must be modified and the text regarding this must be modified.
2- Why was PCNA determined by immunohistochemistry and Western blot techniques and others (Ki67, beta catenin, ….) were not.
3- In Western immunoblots, what is meant by total protein; is it a protein of a house keeping gene beta actin or G6PDH?
4- It is recommended to measure active subunits NF-κB p50 and NF-κB p65 if this is possible.
5- I recommend to add schematic figure at the end of discussion to summarize the suggested mechanisms of action of sorafenib and crocin.
Minor revisions is required
Author Response
Reviewer # 3
Comment 1:
Diethylnitrosamine is used in this model of cancer for initiation of carcinogenesis not to induce liver cirrhosis. Thus, the title must be modified and the text regarding this must be modified
Response:
We disagree with you on this point because the use of Diethylnitrosamine in small, frequent doses causes a model of cirrhosis-hepatocellular carcinoma as shown in the research documented references (Schiffer et al., 2005; DePeralta et al., 2016; Velasco-Loyden et al., 2017).
Details are below:
1- In the introduction on page 4 (introduction): lines 186-189
it was stated that “In humans, approximately 80% of advanced HCC develops from cirrhotic livers [35]. Therefore, novel therapeutic approaches should be tested in an animal model of HCC occurring on a cirrhotic background". Therefore, a rat model of DEN-induced liver injury that replicates the evolution of cirrhosis toward HCC was developed for the current study [36]. In this animal model, Multiple smaller doses of DEN are used in this procedure to cause HCC with a cirrhotic background. This is implemented to mimic the progression of liver cancer in HCC patients by simulating the inflammatory conditions from which fibrosis and cirrhosis arise before malignant tumors are formed [37].
2- In the results, it was confirmed that the incidence of cancer in this animal model was accompanied by liver cancer inflammation and fibrosis, as shown in Figure 4.
References in manuscript
- Velasco-Loyden, G.; Perez-Martinez, L.; Vidrio-Gomez, S.; Perez-Carreon, J. I.; Chagoya de Sanchez, V., Cancer chemoprevention by an adenosine derivative in a model of cirrhosis-hepatocellular carcinoma induced by diethylnitrosamine in rats. Tumour Biol 2017, 39, (2).
- Schiffer, E.; Housset, C.; Cacheux, W.; Wendum, D.; Desbois-Mouthon, C.; Rey, C.; Clergue, F.; Poupon, R.; Barbu, V.; Rosmorduc, O., Gefitinib, an EGFR inhibitor, prevents hepatocellular carcinoma development in the rat liver with cirrhosis. Hepatology 2005, 41, (2), 307-14.
- DePeralta, D. K.; Wei, L.; Ghoshal, S.; Schmidt, B.; Lauwers, G. Y.; Lanuti, M.; Chung, R. T.; Tanabe, K. K.; Fuchs, B. C., Metformin prevents hepatocellular carcinoma development by suppressing hepatic progenitor cell activation in a rat model of cirrhosis. Cancer 2016, 122, (8), 1216-27.
Comment 2:
Why was PCNA determined by immunohistochemistry and Western blot techniques and others (Ki67, beta catenin, ….) were not.
Response:
I would like to explain to you that this study is a master's thesis, and unfortunately, the lack of capabilities was an obstacle that prevented the ability to estimate all indicators with both immunohistochemistry and Western blot techniques, but the student was able to study PCNA with the two techniques because it was available.
Comment 3:
In Western immunoblots, what is meant by total protein; is it a protein of a house keeping gene beta actin or G6PDH?
Response:
1- Housekeeping proteins are essential internal controls for normalization as they are expected to be stably expressed. But there is a problem with the stability of the expression level of housekeeping-specific proteins in hepatocellular carcinoma samples. We used several housekeeping proteins such as ACT, GAPDH and TUBB, but we found housekeeping proteins showed both increases and decreases in tumor tissues, the variations of the different expression levels in the same tissue and group decreased the statistical power of the overall evaluations. We concluded that housekeeping proteins do not “keep house” and that an alternative endogenous control is in need to eliminate the errors from sample amounts. There is research confirming that the direct measurement of total protein amounts exhibited an excellent performance (Hu et al., 2016). The band intensities of total protein staining with SYPRO Ruby exhibited a strong linear correlation with the actual loading amounts, offering a preferable choice to present the total protein amount and serve as a good normalizer.
2- On page seven from lines 360-366: it is mentioned how total protein was utilized as the internal control Total protein was utilized as the internal control and was stacked into a single band on a 5% SDS-PAGE stacking gel with no separating gels. SYPRO Ruby protein gel stain was utilized for total protein staining in accordance with the manufacturer's (#170-3125, BIO-RAD, Hercules, CA, USA) protocol [43]. The band intensities were quantified using ImageJ software (National Institutes of Health).
Reference
- Hu, X.; Du, S.; Yu, J.; Yang, X.; Yang, C.; Zhou, D.; Wang, Q.; Qin, S.; Yan, X.; He, L.; Han, D.; Wan, C., Common housekeeping proteins are upregulated in colorectal adenocarcinoma and hepatocellular carcinoma, making the total protein a better "housekeeper". Oncotarget 2016, 7, (41), 66679-66688.
Comment 4:
It is recommended to measure active subunits NF-κB p50 and NF-κBp65 if this is possible
Response:
Thank you very much for your useful suggestion. We will study active subunits NF-κB p50 and NF-κBp65 in a future study on liver cancer cells, but we now apologize for conducting this study with our current research due to the lack of tissues and capabilities.
Comment 5:
I recommend to add schematic figure at the end of discussion to summarize the suggested mechanisms of action of sorafenib and crocin.
Response:
Thank you very much for your useful recommendation. We have now added Figure 8 at the end of the discussion to summarize the suggested mechanisms of action of sorafenib and crocin.
Round 2
Reviewer 1 Report
All my comments on the previous version of the manuscript have been appropriately addressed. Thank you very much.
Author Response
We are so grateful to the respected reviewer for the great insights.
Reviewer 2 Report
Authors modified manuscript according to reviewer suggestions
Author Response

(The authors gave the same response as above.)
